# Generating Mechanism of Catalytic Effect for Hydrogen Absorption/Desorption Reactions in NaAlH4–TiCl3

**Kazutaka Ikeda** [1,2,3,*], **Fumika Fujisaki** [3,†], **Toshiya Otomo** [1,2,3,4], **Hidetoshi Ohshita** [1], **Takashi Honda** [1,2,3], **Toru Kawamata** [5], **Hiroshi Arima** [5,‡], **Kazumasa Sugiyama** [5], **Hitoshi Abe** [1,3,4], **Hyunjeong Kim** [6], **Kouji Sakaki** [6], **Yumiko Nakamura** [6], **Akihiko Machida** [7], **Toyoto Sato** [5,§], **Shigeyuki Takagi** [5] **and Shin-ichi Orimo** [5,8]



1   Institute of Materials Structure Science, High Energy Accelerator Research Organization (KEK), Tokai, Ibaraki 319-1106, Japan; toshiya.otomo@j-parc.jp (T.O.); ohshita@post.kek.jp (H.O.); takhonda@post.kek.jp (T.H.); hitoshi.abe@kek.jp (H.A.)
2   J-PARC Center, High Energy Accelerator Research Organization (KEK), Tokai, Ibaraki 319-1106, Japan
3   School of High Energy Accelerator Science, The Graduate University for Advanced Studies, Tsukuba, Ibaraki 305-0801, Japan; fumika.fujisaki@gmail.com
4   Graduate School of Science and Engineering, Ibaraki University, Mito, Ibaraki 310-8512, Japan
5   Institute for Materials Research, Tohoku University, Sendai, Miyagi 980-8577, Japan; kawamata@tohoku.ac.jp (T.K.); h_arima@cross.or.jp (H.A.); kazumasa@imr.tohoku.ac.jp (K.S.); toyoto@shibaura-it.ac.jp (T.S.); shigeyuki.takagi@imr.tohoku.ac.jp (S.T.); orimo@imr.tohoku.ac.jp (S.-i.O.)
6   National Institute of Advanced Industrial Science and Technology, Tsukuba, Ibaraki 305-8569, Japan; hj.kim@aist.go.jp (H.K.); kouji.sakaki@aist.go.jp (K.S.); yumiko.nakamura@aist.go.jp (Y.N.)
7   Quantum Beam Science Research Directorate, National Institutes for Quantum and Radiological Science and Technology, Sayo, Hyogo 679-5148, Japan; machida.akihiko@qst.go.jp
8   Advanced Institute for Materials Research (WPI-AIMR), Tohoku University, Sendai, Miyagi 980-8577, Japan
*   Correspondence: kikeda@post.j-parc.jp; Tel.: +81-29-284-4639
†   Present address: Innovative Research Excellence, Honda R&D Co., Ltd., Haga, Tochigi 321-3393, Japan.
‡   Present address: Neutron Science and Technology Center, Comprehensive Research Organization for Science and Society (CROSS), Tokai, Ibaraki 319-1106, Japan.
§   Present address: Department of Engineering Science and Mechanics, College of Engineering, Shibaura Institute of Technology, Toyosu, Koto-ku, Tokyo 135-8548, Japan.

**Abstract:** The hydrogen desorption and absorption reactions of the complex metal hydride NaAlH4 are disproportionation processes, and the kinetics can be improved by adding a few mol% of Ti compounds, although the catalytic mechanism, including the location and state of Ti, remains unknown. In this study, we aimed to reveal the generating mechanism of catalytic Al–Ti alloy in NaAlH4 with TiCl3 using quantum multiprobe techniques such as neutron diffraction (ND), synchrotron X-ray diffraction (XRD), anomalous X-ray scattering (AXS), and X-ray absorption fine structure (XAFS). Rietveld refinements of the ND and XRD, profiles before the first desorption of NaAlD(H)4–0.02TiCl3 showed that Al in NaAlD(H)4 was partially substituted by Ti. On the other hand, Ti was not present in NaAlH4, and Al–Ti nanoparticles were detected in the XRD profile after the first re-absorption. This was consistent with the AXS and XAFS results. It is suggested that the substitution promotes the formation of a highly dispersed nanosized Al–Ti alloy during the first desorption process and that the effectiveness of TiCl3 as an additive can be attributed to the dispersion of Ti.

**Keywords:** neutron diffraction; X-ray diffraction; anomalous X-ray scattering; X-ray absorption fine structure; hydrogen storage; hydride complex

## 1. Introduction

Hydrogen is considered an energy carrier as it is abundant and has a low environmental load. Safe and efficient hydrogen storage techniques remain a significant challenge in the coming hydrogen energy society. The main scientific concern is on the development of hydrogen storage materials, including metal/alloy hydrides, complex hydrides,

ammonia borane, and metal–organic frameworks, which should meet the requirements of high storage capacity, appropriate thermodynamic properties, reversibility, and fast adsorption and desorption kinetics [1,2]. Recently, complex hydrides have offered the possibility of designing a potential hydrogen storage system owing to their low weight and high hydrogen density. Complex hydrides are a group of materials that are a combination of hydrogen and group I and II salts of $[AlH_4]^-$, $[BH_4]^-$, and $[NH_2]^-$, that is, alanates, borohydrides, and amides [3].

$NaAlH_4$ is a complex hydride [4]. Its crystal structure has been observed using single-crystal X-ray diffraction [5], and a space group of $I4_1/a$ with $a = 5.0119(1)$ and $c = 11.3147(5)$ Å at 295 K has been determined for $NAlD_4$ by powder neutron diffraction [6]. The Na atoms in $NaAlD_4$ are surrounded by eight D atoms from eight different $[AlD_4]^-$ tetrahedra in the geometry of a distorted square antiprism. $NaAlH_4$ decomposes in the following three-step disproportionation process with a hydrogen desorption reaction [7]:

$$NaAlH_4 \;\rightleftarrows\; \frac{1}{3}Na_3AlH_6 + \frac{2}{3}Al + H_2 \tag{1}$$

$$\frac{1}{3}Na_3AlH_6 \;\rightleftarrows\; NaH + \frac{1}{3}Al + \frac{1}{2}H_2 \tag{2}$$

$$NaH \;\rightarrow\; Na + \frac{1}{2}H_2 \tag{3}$$

Half of the hydrogen in $NaAlH_4$ (hydrogen-to-metal ratio; $\Delta H/M = 1$; 3.7 mass%) is attributed to Reaction (1) and a quarter ($\Delta H/M = 0.5$; 1.9 mass%) to Reaction (2). The melting point of $NaAlH_4$ is 454 K, and $Na_3AlH_6$ decomposes at 525 K [8]. Reaction (3) ($\Delta H/M = 0.5$; 1.9 mass%) requires a temperature above 1000 K; Reactions (1) and (2) are important for hydrogen storage properties. Because all the reactions are sluggish, Reactions (1) and (2) are sometimes considered irreversible.

Bogdanović and Schwickardi demonstrated that doping the complex metal hydride $NaAlH_4$ with a few mol% of β-$TiCl_3$ or $Ti(OBu)_4$ lowers the decomposition temperature, improves the kinetics, and, importantly, allows re-absorption of hydrogen for the decomposition products (recombination) [9]. In the system of $NaAlH_4$ with Ti compounds, the above two-step reaction proceeds below the melting points [10]. The subsequent hydrogen desorption behavior of $TiCl_3$-doped $NaAlH_4$ seems to reproduce the first hydrogen desorption ($\Delta H/M \sim 1.25$; 4.6 mass%), except for the lower reversible hydrogen capacity ($\Delta H/M = 1.0$–1.1; 3.8–4.0 mass%) [10,11]. Improvements in the kinetics of hydrogen desorption/absorption slowed with increasing amounts of added $TiCl_3$. However, an Arrhenius analysis of the hydrogen desorption indicated that there is a significant drop in the activation energy of the decomposition reaction, $E_a$, upon the addition of $TiCl_3$ (−40% and −20% for $NaAlH_4$ and $Na_3AlH_6$, respectively, with the addition of 0.9 mol% of $TiCl_3$), which then remains approximately constant for higher doping levels [12]. Only the Arrhenius pre-exponential factor was $TiCl_3$-concentration dependent. This suggests that the presence of Ti is a starting point for the recombination reaction of the undoped material.

After the first report on Ti-doped $NaAlH_4$, other Ti-source materials have been explored as dopants. Sandrock et al. pointed out that the $Ti(OBu)_4/Zr(OPr)_4$ alkoxide mechanical doped materials have low reversible capacities and release significant levels of hydrocarbon impurities [13]. Many types of Ti compounds, such as $TiCl_4$, TiN, $TiF_3$, $TiH_2$, and $TiO_2$, have shown this effect on some level [14–22]; however, $TiCl_3$ was found to show promising results [11]. The following reaction is assumed to proceed when $TiCl_3$ is added to $NaAlH_4$:

$$NaAlH_4 + xTiCl_3 \;\rightarrow\; (1-3x)NaAlH_4 + 3xNaCl + 3xAl + xTi + 6xH_2 \;(x < 1/3) \tag{4}$$

During this reaction, Al and Ti may form a stable Al–Ti alloy phase.

The position of Ti in the doped material and the oxidation state of the Ti after doping remain highly controversial despite many studies and discussions so far [18]. Some com-

putational works insist that Ti atoms or ions generally favor substitution over interstitial sites [23,24], although no conclusion has been reached. This is attributed to the fact that the by-products vary depending on the species of the additives, and the phases involved in the disproportionation reaction are complicated. On the other hand, a common observation is the formation of an Al–Ti alloy after the hydrogen desorption/absorption cycles [25–29], and Ti compounds are not detected in the diffraction measurements immediately after the addition [25,26,30,31]. It is suggested that the formation of $TiAl_3$ upon doping with $TiCl_3$ contributes to enhanced kinetics [32]. In fact, nanocrystalline $Ti_3Al$ with a longer milling time exhibits a catalytic function in the hydrogen re-absorption/desorption reaction of $NaAlH_4$ [33]. However, the direct addition of $TiAl_3$ or $Ti_3Al$ alloys to $NaAlH_4$ does not yield effective catalysis [14], and Ti-doped $NaAlH_4$ after 100 cycles includes additional phases that are apparently linked to decreased hydrogen desorption kinetics and total amount of released hydrogen [34]. These results imply that the microstructure of the Ti compound significantly contributes to the catalytic effect on $NaAlH_4$.

A structural analysis of $Ti(OBu)_4$-doped $NaAlH_4$ was performed using powder X-ray diffraction [25]. No phases other than $NaAlH_4$ were detected immediately after the addition, although $NaAlH_4$, $Na_3AlH_6$, Al, NaH, and a small amount of unidentified phases were observed after hydrogen storage cycles. A shoulder on the high scattering angle side of the main Al diffraction peak (111) was noticed, the magnitude of which increased with the amount of the applied catalyst and which could be tentatively assigned to the Al–Ti alloy. The composition of the Al–Ti phase was estimated to be $Al_{0.85}Ti_{0.15}$ [35] or $Al_{0.5}Ti_{0.5}$ [36]. The formation of this phase may explain the reduction in capacity beyond the theoretical reduction from the dead weight of the additive and the reaction of $Na_3AlH_6$ with Al and hydrogen to $NaAlH_4$ [25]. In addition, an extended X-ray absorption fine structure (EXAFS) analysis indicated the formation of $TiAl_x$ nanoparticles such as $TiAl_3$ [37,38]. The possibility of Ti substitution into the $NaAlH_4$ host has been reported [30,39], and it is implied that Ti is substituted at the Al site in $NaAlH_4$ [31,40] or in $Na_3AlH_6$ [41]. However, the solubility of Ti in $NaAlH_4$ is sometimes denied because of the lack of lattice parameter changes in the Rietveld refinement result [28].

In this study, the changes in the atomic arrangement and chemical state during the hydrogen desorption/re-absorption process of a $NaAlH_4$–$TiCl_3$ system were investigated to elucidate the mechanism of the reversible disproportionation reactions by structural analysis using neutron diffraction (ND), X-ray diffraction (XRD), anomalous X-ray scattering (AXS), and X-ray absorption fine structure (XAFS). We determined how the position and oxidation state of Ti in this system changed through the reactions and how they contribute to the hydrogen storage characteristics.

## 2. Materials and Methods

The direct dry synthesis of $NaAlH_4$ has been explained in a previous publication [42]. $NaAlD_4$ was similarly prepared using NaD (Japan Metals & Chemicals Co., Ltd., 1-17-25, Shinkawa, Chuo-ku, Tokyo 104-8257, Japan) and $AlD_3$ [43] instead of NaH and $AlH_3$ [44], respectively. $NaAlD_4$ or $NaAlH_4$ (Merck KGaA, Darmstadt, Germany, 93% pure) and $TiCl_3$ (Merck KGaA, Darmstadt, Germany, 99.995+% pure) were mixed to obtain the composition of $NaAlD(H)_4$–$xTiCl_3$ ($x$ = 0–0.06). Each mixture (500 mg) and 20 steel balls with a diameter of 7 mm were placed in a steel vial with a volume of 30 $cm^3$ and mechanically milled in a planetary ball mill apparatus (Fritsch P7) at 400 rpm for 3 h at ambient temperature. The pressure–composition (*p–c*) isotherms of $NaAlD_4$–$0.02TiCl_3$ were measured at temperatures ranging from 398 to 408 K. The desorption measurement of the as-milled sample under 10 MPa $D_2$ at 403 K was started first, and then the sample vessel was evacuated before measuring every absorption up to 10 MPa $D_2$. The as-milled sample was heat-treated for 10 h at 9.5 MPa of deuterium or hydrogen at 403 K to obtain the neutron/X-ray diffraction patterns of the samples with larger crystallite sizes and fewer lattice strains. To minimize (hydro-)oxidation or nitrogenation, the samples were handled in a glove box filled with purified helium of less than 1 ppm oxygen and a dew point at a temperature below 183 K.

The averaged structure of the samples (200 mg) in a V-Ni null scattering sample container with an outer diameter of 6.0 mm and a thickness of 0.1 mm were examined before and after the hydrogen desorption/absorption for an exposure time of 4 h at room temperature. This was done by ex-situ neutron diffraction on a neutron total scattering spectrometer (NOVA) (beamline BL21, with a decoupled liquid hydrogen moderator, an incident flight path of 15 m, and a scattered flight path of 1.0–1.4 m at the backscattering detector bank ($0.06 \leq d \leq 5.2$ Å) connected to a 300 kW spallation neutron source at the Japan Proton Accelerator Research Complex (J-PARC). Scattering data were collected over a lattice spacing range of 0.1–5.2 Å for neutron diffraction at room temperature. The consistency in the scattering data was confirmed using standard materials such as silicon powder (NIST SRM 640d). Synchrotron radiation X-ray diffraction measurements were carried out using a Rigaku R-AXIS V detector at the beamline BL22XU at SPring-8. Scattering data were collected over a lattice spacing range of 0.3–6.5 Å for X-ray diffraction with an incident energy of 70.21 keV (wavelength of 0.1766 Å) at room temperature. The same samples were placed in a polyimide film tube with an inner diameter of 1 mm and a thickness of 0.05 mm and sealed with epoxy resin on both sides. The averaged crystal structure refinements over a lattice spacing range of 0.6–5.0 Å for neutron diffraction at a backscattering detector bank and 0.8–5.0 Å for X-ray diffraction, respectively, were performed by Rietveld refinement using the computer program Z-Rietveld [45,46].

Anomalous X-ray scattering was performed with an ionization chamber monitor and an Ortec IGLET-16160 detector (Ge SSD) at BL7C at the Photon Factory, KEK. The gas path was equipped around the sample and between the monitor and the sample holder filled with He. The Ge SSD was placed immediately behind the sample through a solar slit. The incident energy was calibrated using a Cu film (8.9803 keV). The atomic scattering factor of the X-ray can be expressed as:

$$f = f_0(Q) + f'(E) + i f''(E) \tag{5}$$

where $f_0$ is the Thomson scattering term, and $f'$ and $f''$ are anomalous scattering factors [47]. $f''$ is related to absorption. $f'$ and $f''$ are constants, and $f$ is a function of the scattering angle (momentum transfer; $Q$ $(= 2\pi/d = 4\pi\sin\theta/\lambda)$) when the X-ray energy is far from the absorption edge, though $f$ changes with $f'$ and $f''$ depending on the energy because of resonant absorption when the energy is near excitation. Typically, the absolute intensity is measured to evaluate the occupation ratio by Rietveld refinement [48]. However, it is difficult for the Ti K-edge (approximately 4.9 keV) to use the Rietveld method because of unignorable absorption in air and fewer Bragg peaks due to the limited $Q$ range. Thus, the relative intensity was evaluated to identify the substitution site of Ti in $NaAlH_4$.

The Ti K-edge X-ray absorption fine structure was measured at BL9A at the Photon Factory, KEK. Because the amount of Ti is small, the spectra were measured in fluorescence modes to compensate for the low signal-to-noise ratio. The samples were sealed in a polyethylene bag in a glove box. A line was regressed to the obtained data in the pre-edge range (–150 to –30 eV from the absorption edge), and a spline was regressed to the data in the post-edge range (150 to 990 eV). The differences were normalized using the software Athena [49]. The absorption edge of $NaAlH_4$–$TiCl_3$ was determined to be 0.5 by normalization of the range higher than 4.9965 keV (Ti calibrated edge). The edge shift was determined from the oxidation states of the standard samples (Ti: 0, TiO: 2, $Ti_2O_3$: 3, $TiO_2$: 4), as shown in Figure S2. The EXAFS spectrum, $\chi$, against the wavenumber k can be expressed as:

$$\chi(k) = S_0^2 \sum_j \frac{N_j F_j(k) exp\left(-2k^2\sigma_j^2\right)}{kr_j^2} \sin\left(2kr_j + \phi_j(k)\right) \tag{6}$$

where $S_0^2$ is an intrinsic loss factor, $N_j$ is the coordination number, $F_j(k)$ is the backscattering factor, $\sigma_j$ is the Debye–Waller factor, $\phi_j$ is the phase factor, and $r$ is the interatomic distance.

$S_0^2$ was determined to be 0.61 using a standard Ti film sample. The radial distribution functions obtained from the EXAFS spectra were fitted using the software Artemis [49].

The ground-state (0 K) thermodynamic stability of Na(Al, Ti)H$_4$ was investigated using first-principles density functional theory (DFT) calculations as implemented in the Vienna ab initio simulation package (VASP) [50,51] with a plane-wave basis and the projector augmented wave method [52,53] within the generalized gradient approximation with the Perdew–Burke–Ernzerhof exchange-correlation functional [54]. A cutoff energy of 780 eV and a Gaussian smearing method with an energy broadening of 0.2 eV were used throughout. The k-mesh generated using the Monkhorst-Pack method had a spacing of less than 0.5 Å$^{-1}$. The criterion for self-consistency in the electronic structure determination was that two consecutive energies differed by less than 0.1 meV. The ground-state energetics of the elements and compounds in their most stable standard state phase, Na metal ($Im\bar{3}m$), Al metal ($Fm\bar{3}m$), Ti metal ($P6_3/mmc$), and H$_2$ molecule gas ($15 \times 15 \times 15$ Å$^3$, nonspin-polarized) were used to determine the thermodynamic changes for the NaAlH$_4$ ($I4_1/a$) doping reactions. The ground state geometries of Na(Al, Ti)H$_4$ were determined by minimizing the Hellman–Feynman forces with the residual minimization method-direct inversion in the iterative subspace (RMM-DIIS) implementation of the quasi-Newton method. Full minimization of the atomic coordinates, cell size, and cell shape was conducted simultaneously. The structure was relaxed when all the forces of the nonfixed atoms were <0.05 eV/Å.

## 3. Results

### 3.1. Pressure–Composition Isotherm (PCI)

The PCI curves of the NaAlD$_4$–0.02TiCl$_3$ were measured at 398, 403, and 408 K, as shown in Figure 1. Two plateau pressures can be clearly observed in the deuterium desorption reactions at each temperature, indicating the deuterium desorption reactions similar to that observed in the NaAlH$_4$–TiCl$_3$ system. The enthalpy change, $\Delta H$, in Reactions (1) and (2), was calculated by using the van 't Hoff plot. The $\Delta H_{(1)}$ and $\Delta H_{(2)}$ values are $-32.8$ and $-54.2$ kJ/mol D$_2$ for Reactions (1) and (2), respectively. The values are comparable to the reported values of $-37$ and $-47$ kJ/mol H$_2$ for NaAlH$_4$ and Na$_3$AlH$_6$, respectively [10]. The hydrogen desorption amount was decreased from $\Delta D/M = 1.24$ for the first desorption to $\Delta D/M = 1.04$–1.12 after the subsequent one, similar to the previous reports [10,11].

### 3.2. Neutron/X-ray Diffraction (ND/XRD)

The neutron powder diffraction profiles of NaAlD$_4$ and NaAlD$_4$–0.02TiCl$_3$ before the desorption and after the re-absorption of deuterium were measured at NOVA, as indicated by the circles in Figure 2a. In the plotted profile, the contaminations of the background intensities from the sample cell and the spectrometer were subtracted, and the neutron attenuation factors of the sample and sample cell were calibrated. Figure 2b shows the synchrotron X-ray powder diffraction profiles. Table 1 represents the Rietveld refinement results.

The deuteride sample without TiCl$_3$ includes a small amount of Na$_3$AlD$_6$ in the neutron and X-ray diffraction profiles; nevertheless, the purity is high enough (97%) compared to a previous report [6]. The TiCl$_3$-doped samples before the desorption were composed of NaAlD$_4$, NaCl, and Al in both the diffraction profiles. The Rietveld refinement results implied that Ti was substituted at approximately 2% of the Al sites in NaAlD$_4$. The unit cell volume change is not significant from the substitution despite the difference in the effective ionic radii with four coordination (Al$^{3+}$: 0.39 Å and Ti$^{4+}$: 0.42 Å) [55]. On the other hand, the phases of NaAlD$_4$, Na$_3$AlD$_6$, NaCl, and Al were identified in the neutron diffraction profile for the sample after the re-absorption; however, Na$_3$AlD$_6$ was very small in the X-ray profile because of its low symmetry and small amount. The amount of NaAlD$_4$ decreased from 4% (neutron) and 10% (X-ray) by one-time desorption and re-absorption, whereas the *p-c* isotherms showed 13%. Although the results are the same in terms of the reduction tendency, it is difficult to make a quantitative evaluation.

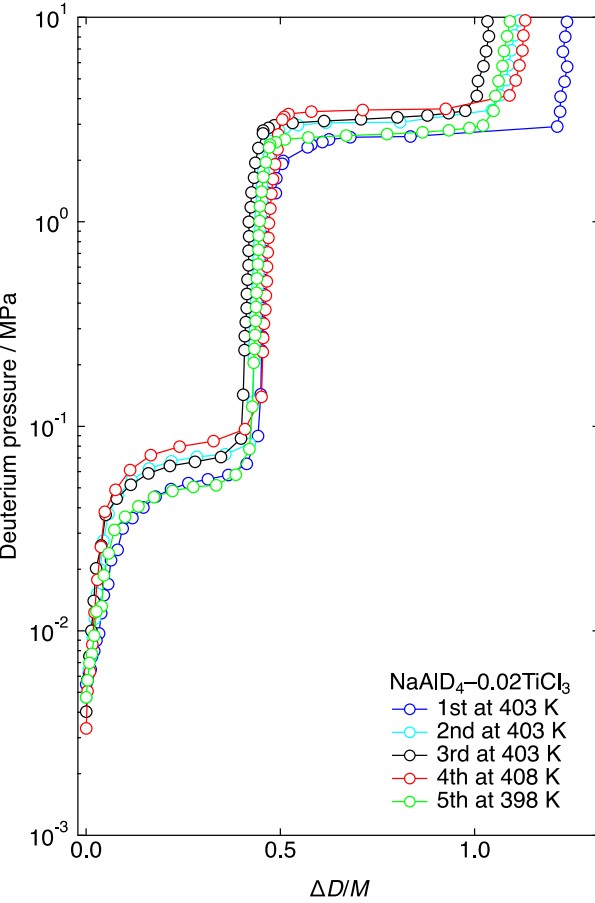

**Figure 1.** Hydrogen pressure–composition (*p*–*c*) isotherm of deuterium desorption of NaAlD$_4$–0.02TiCl$_3$ at 398, 403, and 408 K.

**Table 1.** Summary of Rietveld refinement of NaAlD$_4$ and NaAlD$_4$–0.02TiCl$_3$ before the desorption and after the re-absorption of deuterium.

| Comp. | Condition | Method | Unit Cell Volume/Å³ | Occupancy at Al | | Phase Ratio | | | | Reliability |
|---|---|---|---|---|---|---|---|---|---|---|
| | | | | Al | Ti | NaAlD$_4$ | Na$_3$AlD$_6$ | NaCl | Al | $R_{wp}$/% |
| NaAlD$_4$ | before dope | neutron | 285.608 (4) | 1.0 | 0 | 0.9723 (10) | 0.0276 (3) | - | 0.0000 (19) | 2.36 |
| | | X-ray | 284.864 (9) | 1.0 | 0 | 0.9753 (13) | 0.0063 (4) | - | 0.0183 (14) | 2.51 |
| NaAlD$_4$– 0.02TiCl$_3$ | before 1st des. | neutron | 285.640 (3) | 0.9786 (5) | 0.0213 (5) | 0.9019 (5) | - | 0.0665 (19) | 0.0314 (12) | 4.24 |
| | | X-ray | 284.385 (5) | 0.97 (2) | 0.02 (2) | 0.862 (4) | - | 0.061 (2) | 0.076 (3) | 1.98 |
| | after 1st re-abs. | neutron | 285.629 (16) | 0.9945 (5) | 0.0054 (5) | 0.8663 (4) | 0.03931 (16) | 0.07328 (18) | 0.0210 (8) | 5.64 |
| | | X-ray | 284.703 (6) | 0.99 (2) | 0.00 (2) | 0.777 (3) | 0.00179 (18) | 0.0581 (13) | 0.162 (3) | 2.34 |

　　　　In the sample after the re-absorption of deuterium, there was hardly any Ti at the Al site in NaAlD$_4$. The phase ratio of Al after the re-absorption was 2.1 times as much as that before the desorption in the X-ray diffraction profile, and approximately 4% of Na$_3$AlD$_6$ appeared in the neutron diffraction. Compared with before the desorption and after the re-absorption in the X-ray diffraction profiles, a new peak appeared at the low scattering angle (long lattice spacing, *d*) side of the 111 Bragg peak (*d* = 2.3 Å) of Al after the reaction, but there is no change in the 200 peak (*d* = 2.0 Å), as shown in Figure S1. Brinks et al. insisted that an Al-rich Al–Ti intermetallic phase of fcc with small substitutions of Ti (on the low scattering angle side of the 111 and 200 peaks of Al) is formed depending on the additives of 10% TiCl$_3$ and 2% Ti(OBu)$_4$ after prolonged cycles of desorption at 433 K and absorption at 393 K, respectively [35]. There is a possibility that cycling at different temperatures could result in different compositions of Al–Ti (different Bragg peak positions and relative intensities in the diffraction profile).

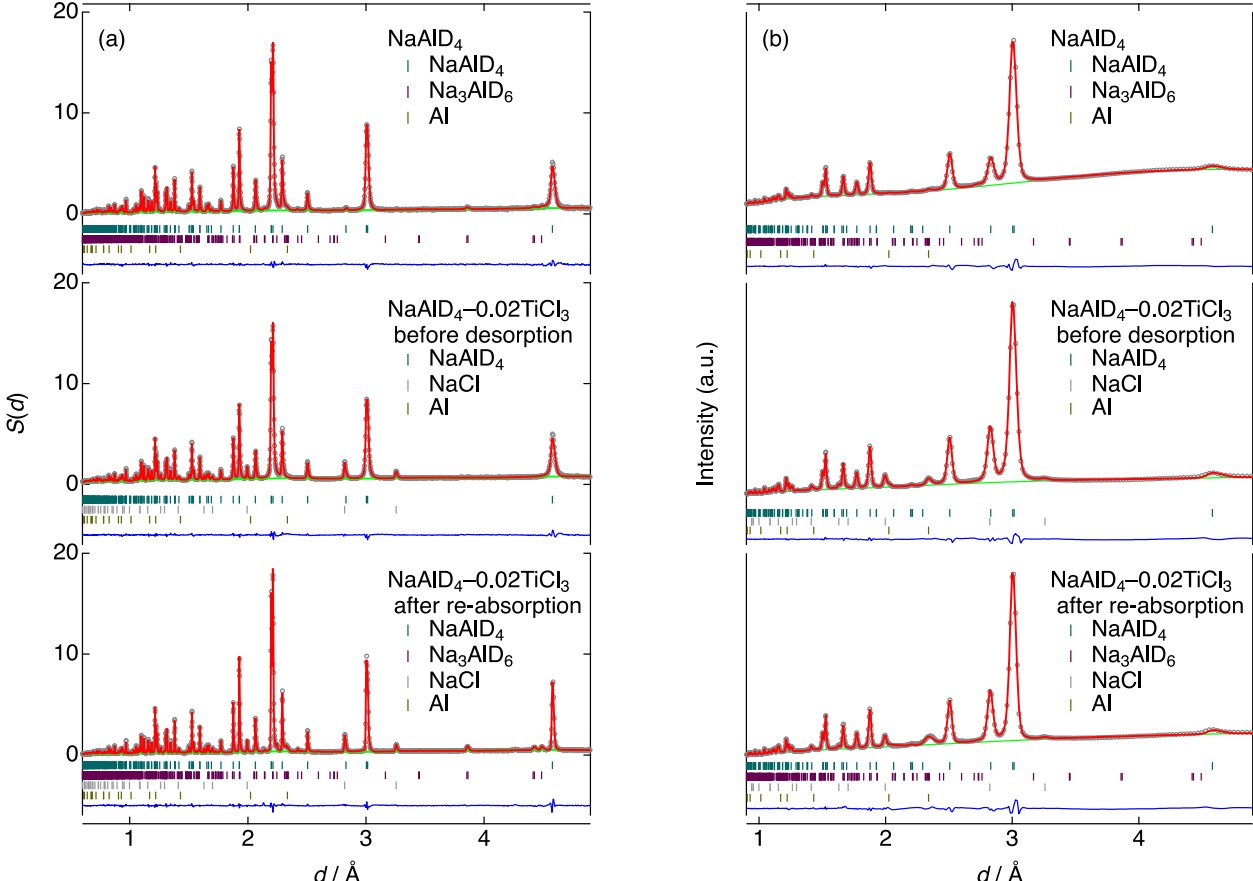

**Figure 2.** Powder diffraction profiles of NaAlD$_4$ and NaAlD$_4$–0.02TiCl$_3$ before the desorption and after the re-absorption of deuterium by (**a**) neutron and (**b**) synchrotron X-ray. Rietveld refinement results: observed (circles), calculated (line), and residual (line below the vertical bars) diffraction profiles. The Bragg reflection positions are shown for NaAlD$_4$, Na$_3$AlD$_6$, NaCl, and Al.

### 3.3. Anomalous X-ray Scattering (AXS)

The crystal structure factors of the 101 and 112 planes of NaAlH$_4$ can be expressed as:

$$F^{101} = 2\sqrt{2}(f_{Na} - f_{Al}) - 2.4f_H \tag{7}$$

$$F^{112} = -4(f_{Na} + f_{Al}) - 0.62f_H \tag{8}$$

The crystal structure factors have different energy dependences in the case of Na and Al substituted by Ti. The intensity of 101 decreases by approximately 3%, and that of 112 is almost constant for the 2% Ti-substituted Al. On the other hand, the intensity of 101 increases for the Ti-substituted Na. Therefore, the relative intensity of 101/112 is available to identify the metal sites that are substituted by Ti.

The diffraction peak intensities for 101 and 112 of NaAlH$_4$–0.02TiCl$_3$ were measured using incident X-ray energies of 4.6645 and 4.9395 keV, respectively. As the desorption reaction gradually progressed even at room temperature, all the peak intensities without the degradation were summed for alternately repeating measurements of the two diffraction peaks. Figure S3 shows the background with interpolation by linear function and integration range of $d$ = 4.55 $\pm$ 0.07 and 3.005 $\pm$ 0.035 Å for 101 and 112 of each Bragg peak.

The relative intensities of the 101 and 112 peaks for different incident X-ray energies, $I^{101/112}_{E/4.6645\ keV}$, were evaluated by simulation, as shown in Figure 3. $I^{101/112}_{4.9395\ keV/4.6645\ keV}$ is 0.962 $\pm$ 0.0097, and the reduction of 3.8% is significant because of the relatively low statistical error (0.82%). The simulation curves show the three cases of Ti-substituted

Al, Ti-substituted Na, and nonsubstituted (interstitial Ti) in Figure 3. By comparing $I_{E/4.6645\ keV}^{101/112}$ with the simulation curves, we found that Ti clearly occupies the Al site in NaAlH$_4$. Additionally, the relationship between the relative intensity and the amount of Ti substitution at the Al site was estimated, as shown in Figure 4. This result shows that $1.3 \pm 0.3\%$ of Ti is present at the Al site in NaAlH$_4$. The X-ray absorption edge shift should be considered to obtain an accurate substitution amount based on TiO$_2$, although it is difficult to determine the edge of Ti at the Al site in NaAlH$_4$. Typically, the oxidation state of Ti in compounds is higher than that in metallic Ti; hence $E_0$ shifts to the high-energy side. In the case of the high-energy-shifted simulation, the substitution amount is increased by more than 1.3%, which is consistent with the Rietveld refinement result before the hydrogen desorption (Table 1).

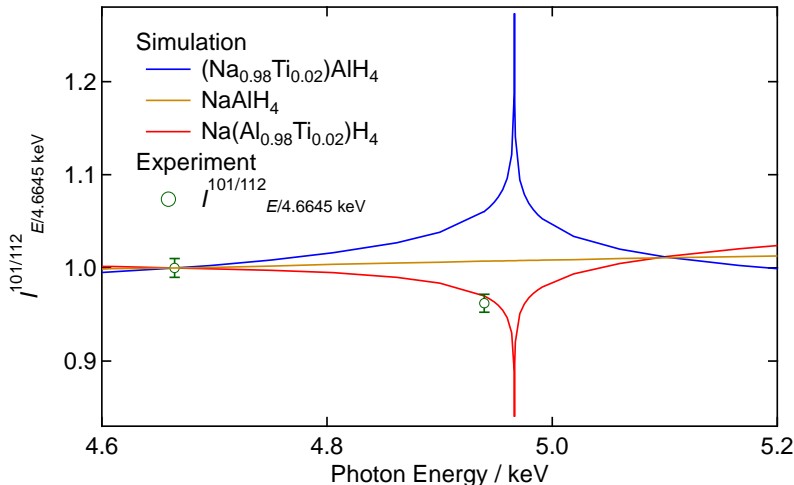

**Figure 3.** Relative intensity obtained from anomalous X-ray scattering measurement of NaAlH$_4$–0.02TiCl$_3$ before the desorption. Simulation curves represent the Ti-substituted Al, Ti-substituted Na, and nonsubstituted cases.

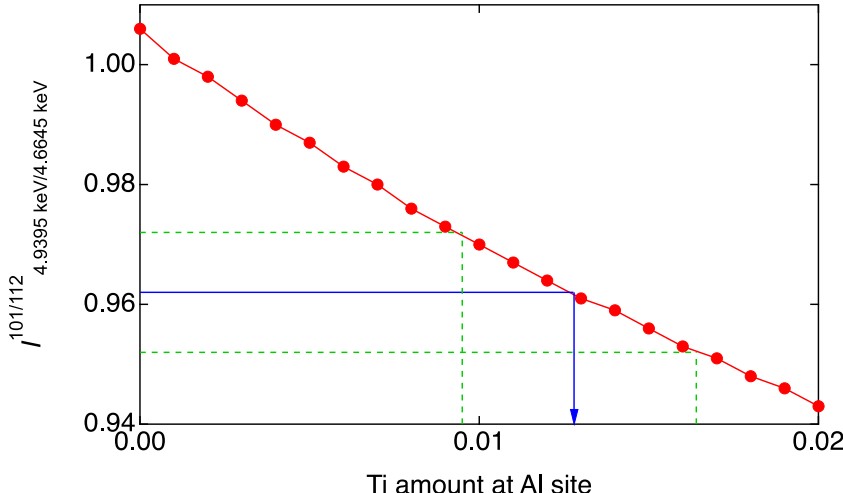

**Figure 4.** Relationship between the relative intensity and the amount of Ti substituted at Al sites. The solid bule line represents the Ti amount at the Al site in NaAlH$_4$ with statistical error (dotted green lines).

### 3.4. Ti K-Edge X-ray Absorption Fine Structure (XAFS)

The oxidation state and local structure of Ti in NaAlH$_4$–$x$TiCl$_3$ ($x$ = 0.005, 0.010, 0.015, 0.020, 0.040, and 0.060) before the desorption and after the re-absorption of hydrogen

were investigated by Ti K-edge X-ray absorption fine structure with reference to standard samples of Ti, TiO, $Ti_2O_3$, $TiO_2$, $TiH_2$, $TiCl_3$, TiAl, $Ti_3Al$, and $TiAl_3$. Figure 5a shows the X-ray absorption near-edge structure (XANES) profiles of $NaAlH_4$–$0.02TiCl_3$ before the desorption and Ti, $TiAl_3$, $TiH_2$, and $TiCl_3$. The XANES spectrum of Ti in $NaAlH_4$–$0.02TiCl_3$ before the desorption was different from that of any of the standard samples. The Ti oxidation number is in the range of 2.3–2.5, as listed in Table S1. Hence, it can be inferred that the chemical state and local structure of Ti in $NaAlH_4$–$0.02TiCl_3$ are independent of the amount of $TiCl_3$.

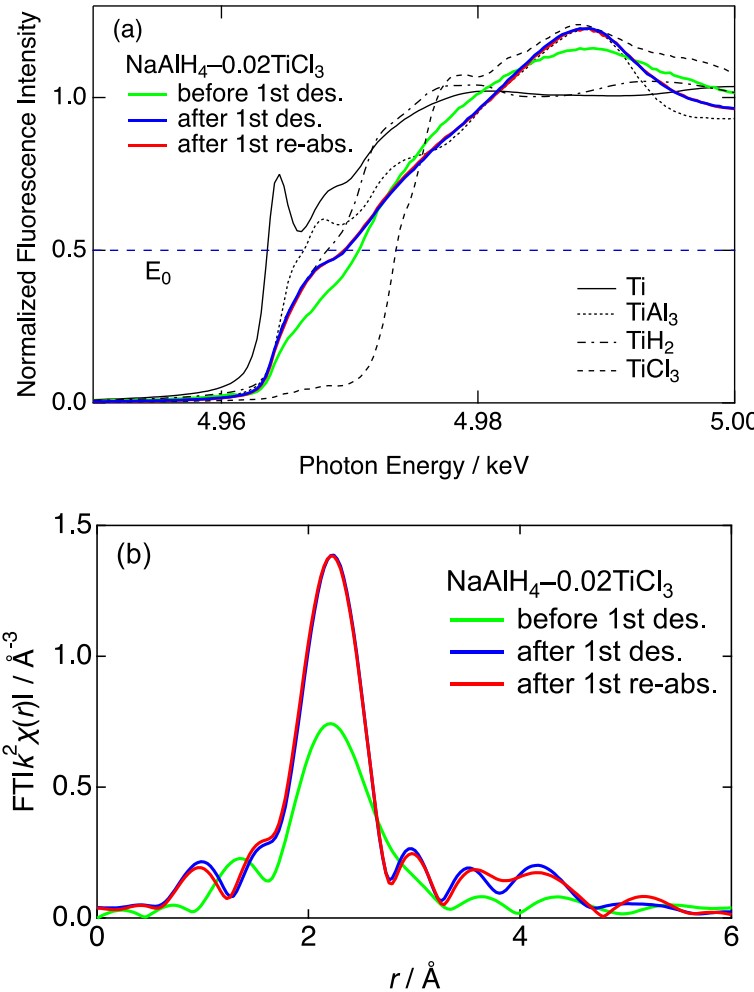

**Figure 5.** (**a**) XANES spectra and (**b**) radial distribution functions of $NaAlH_4$–$0.02TiCl_3$ before the desorption, after the desorption, and after the re-absorption of hydrogen.

The local structure around Ti in $NaAlH_4$–$0.02TiCl_3$ before the desorption was analyzed for radial distribution functions obtained from the EXAFS spectra (green line in Figure 5b) using the three cluster models shown in Table 2. Model A is a single phase of Al–Ti alloy with an intrinsic loss factor, $S_0^2 = 0.61$, which was determined from the measurement of the Ti foil. In model B, the reduction factor was treated as a free parameter for the same Al–Ti alloy. Model C is composed of two phases of Al–Ti alloy and $NaAlH_4$ with substitution of Ti at the Al site, in which $S_0^2 = 0.61 \times$ (phase ratio) $\times$ (normalization factor for the coordination number). Here, the presence ratio of Al–Ti and $Na(Al, Ti)H_4$ is 0.7:1.3 because the total amount of Ti is 0.02 and Ti in $Na(Al, Ti)H_4$ is 0.013 from the above AXS result. The $TiH_2$ path was negligible because it was not detected in the XANES and EXAFS spectra. $Na(Al, Ti)H_4$ has some paths, but isotropic expansion and contraction are assumed. Figure 6 shows the fitting analysis results. Model A is inappropriate, with a high reliability factor (17.8%). Model B is slightly improved, but $S_0^2$ is too low compared with that of

the standard sample, indicating lower Ti–Al coordination numbers for the Al–Ti phase than expected. Model C is the best with the lowest reliability factor (1.1%), and the fitting curve is clearly improved for bond lengths longer than $r$ = 2.5 Å. The bond lengths of Ti–Na and Ti–Al in Na(Al, Ti)$H_4$ were 10% shorter than those of the averaged structure of Rietveld refinements; however, the Debye–Waller factors (Na: 0.05(2), Al: 0.007(2) Å$^2$) were consistent with those of (Na: 0.0342(5), Al: 0.0157(3) Å$^2$).

**Table 2.** Fitting results for the radial distribution functions obtained from the EXAFS spectra of NaAl$H_4$–0.02TiCl$_3$ before the desorption and after the re-absorption of hydrogen.

| Condition | Model | Phase | Symmetry | Shell | Bond Length | Debye–Waller | Edge Shift | Intrinsic Loss | Reliability |
|---|---|---|---|---|---|---|---|---|---|
| | | | | | $r$/Å | $\sigma^2$/Å$^2$ | $\Delta E$/eV | $S_0^2$ | $R$/% |
| before des. | A | Al–Ti | $Fm\bar{3}m$ | Al | 2.83 (5) | 0.023 (5) | −0.6 (20) | 0.61 | 17.8 |
| | B | Al–Ti | $Fm\bar{3}m$ | Al | 2.83 (2) | 0.007 (4) | 0.2 (10) | 0.29 (6) | 4.9 |
| | C | Al–Ti | $Fm\bar{3}m$ | Al | 2.81 (1) | 0.0090 (8) | −1.3 (9) | 0.26 (35) | 1.1 |
| | | Na(Al, Ti)$H_4$ | $I4_1/a$ | Na | 3.15 (5) | 0.05 (2) | −4.0 (36) | 0.23 (23) | |
| | | | | Al | 3.37 (5) | 0.007 (2) | | | |
| | | | | Na | 3.37 (5) | 0.05 (2) | | | |
| after re-abs. | - | Al–Ti | $Fm\bar{3}m$ | Al | 2.780 (15) | 0.0140 (14) | −0.8 (8) | 0.61 | 2.2 |

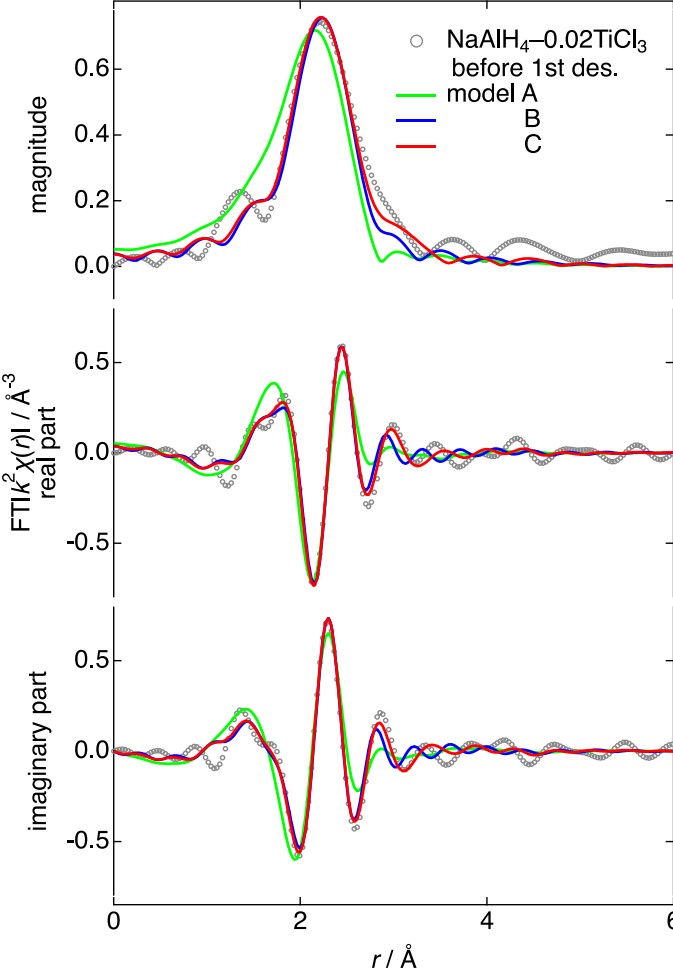

**Figure 6.** Fitting curves for radial distribution functions obtained from EXAFS spectra of NaAl$H_4$–0.02TiCl$_3$ before the hydrogen desorption.

The samples before the desorption and after the re-absorption of hydrogen are mainly composed of $NaAlH_4$, and those after the desorption are composed of NaH and Al, as mentioned above. Figure 5a shows the XANES spectra. The profile after the desorption is different from that before the desorption but almost overlaps with that after the re-absorption. The absorption edges after the desorption and the re-absorption are shifted to the lower energy side (reduced) than that before the desorption, and the oxidation number is 2.0. Figure 5b shows the radial distribution functions before the desorption, after the desorption, and after the re-absorption of hydrogen. The profiles after the desorption and re-absorption almost coincide with each other, similar to the XANES spectra, and the amplitudes are increased and sharpened compared with those before the desorption. Since the first shell peaks in the radial distribution functions after the desorption and re-absorption are similar, it is believed that Ti existing in $NaAlH_4$ never returns after the decomposition. The local structure after the re-absorption was analyzed with the Al–Ti model and fixed $S_0^2 = 0.61$. Figure 7 and Table 2 present the results. The fitting converged to a reliability factor of 2.2%. This Al–Ti alloy was already formed after the desorption because of the similarity of the XAFS profiles with that after the re-absorption.

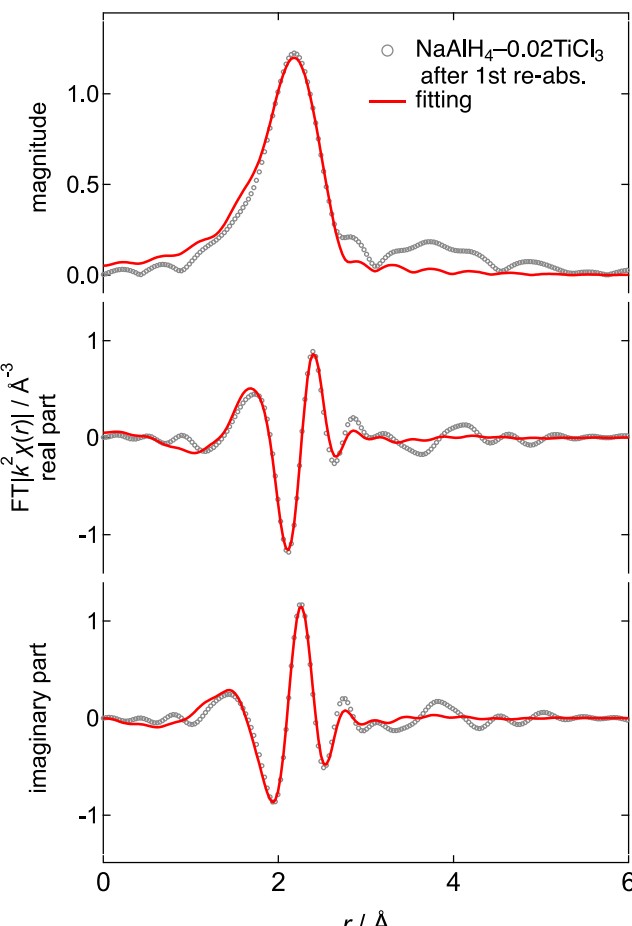

**Figure 7.** Fitting curves for radial distribution functions obtained from the EXAFS spectra of $NaAlH_4$–$0.02TiCl_3$ after the re-absorption of hydrogen.

### 3.5. Density Functional Theory (DFT) Calculations

The location of Ti in doped $NaAlH_4$, based on the assumption that Ti is either replacing one or more atoms in the host lattice or that Ti occupies interstitial sites, has been investigated using density functional theory (DFT) calculations [23]. Løvvik et al. insisted that substitution of Ti for Al, which is closer in ionic radius and formal valence, is more feasible than Ti substitution for a larger, monovalent Na ion, and the most favorable model had Ti

substitution for Al and no vacancy formation, followed by the model with Ti substitution for Al and one H vacancy formed. In the case of adding 3.125–25% Ti to the bulk $NaAlH_4$, the least unfavorable arrangement compared with pure $NaAlH_4$ is Ti replacing Al in the lattice. In this study, the ground state stability of Na(Al, Ti)$H_4$ was investigated using a similar method, based on the results of AXS and XAFS. The enthalpy of formation ($\Delta H_F$) and the heat of substitution ($\Delta H_S$) of the doped compounds were referenced to energies of the stoichiometric equivalents of the constituent atoms in their standard state, as expressed by the following equations:

$$\Delta H_F(NaAl_{1-x}Ti_xH_4) = [E(NaAl_{1-x}Ti_xH_4) - \{E(Na) + (1-x)E(Al) + xE(Ti) + 2E(H_2)\}]/N_{atom} \tag{9}$$

$$\Delta H_S(NaAl_{1-x}Ti_xH_4) = [\{E(NaAl_{1-x}Ti_xH_4) + xE(Al)\} - \{E(NaAlH_4) + xE(Ti)\}]/N_{atom} \tag{10}$$

where $E(Al)$ is the uncorrected total free energy of bulk Al, as calculated using the VASP. All the enthalpies are expressed in kJ/mol divided by the number of atoms $N_{atom}$ (kJ/mol atom) in the substituted periodic cell. The evaluation of the enthalpy changes in the Ti-substituted lattice at a doping level of approximately 1 mol% stoichiometrically required the use of a 768 atom $4 \times 4 \times 2$; 128($NaAlH_4$) supercell constructed from the conventional tetragonal $I4_1/a$; 4($NaAlH_4$) cell. Figure 8 shows the dependences of $\Delta H_F$ and $\Delta H_S$ with doping levels ranging from 0 to 0.125 Ti. Both the formation and substitution enthalpies increased with the amount of Ti. The substitution enthalpy is positive for all the doping levels, indicating that $NaAlH_4$ substituted by the amount of Ti in the AXS result ($x = 0.013$) is metastable.

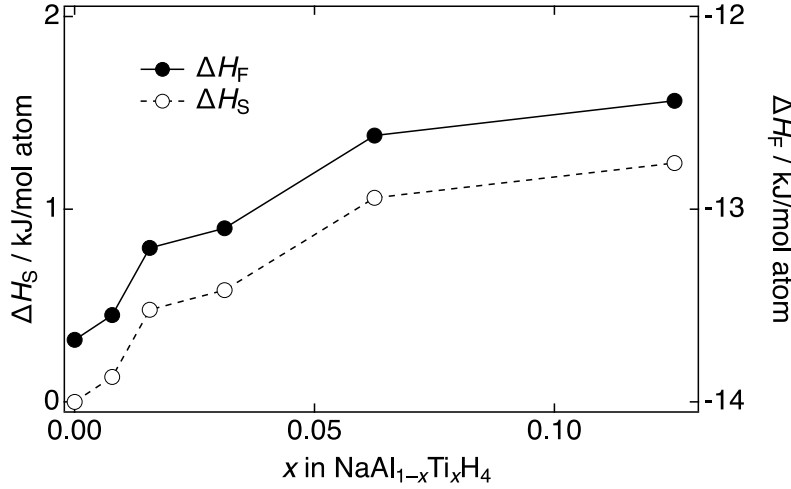

**Figure 8.** Variations of the enthalpy of formation, $\Delta H_F$, and the heat of substitution, $\Delta H_S$, as a function of the Ti content in doped $NaAlH_4$.

## 4. Discussion

The neutron and X-ray diffraction profiles indicate that NaCl and Al form by doping $TiCl_3$ with $NaAlH_4$, according to Equation (4). No Ti-related phase was observed in the diffraction patterns immediately after the addition, whereas Al–Ti alloys with a shoulder on the low scattering angle side of the Al diffraction peak were detected after hydrogen desorption/absorption cycles. This implies the formation of Al–Ti alloys with an fcc structure. However, to detect trace elements, the AXS method is more reliable than typical diffraction measurements because of its elemental selectivity. In this study, the AXS measurements revealed that 1.3% of Ti was substituted at the Al site in $NaAlH_4$ immediately after the addition of $TiCl_3$. So far, substitution of $NaAlH_4$ with Ti has been suggested, but it has not been applied because the lattice constants do not change before and after the addition. Additionally, Ti K-edge XAFS supports that Ti occupies the Al site in $NaAlH_4$ before the hydrogen desorption and that Ti forms Al–Ti alloy. Furthermore, the neutron and X-ray diffraction refinement results suggest the substitution, but the lattice parameter

change was negligible, as in previous reports. The local structure change with shortening of the bond lengths of Ti–Na and Ti–Al derived from the EXAFS analysis may induce a lower symmetry of the crystal structure. Isolated and dispersed nanoparticles of Al–Ti are expected to precipitate because the alloy was not detected in the neutron and X-ray diffraction profiles, and long-range correlation was not present in the EXAFS spectra.

The Ti K-edge EXAFS spectrum after the hydrogen desorption was reproduced with an fcc Al–Ti alloy. This Al–Ti phase originates from the substitution of Ti at the Al site in $NaAlH_4$. Based on the XANES spectra, the oxidation number of Ti was changed from 2.4 to 2.0 through the hydrogen desorption. The similarity in the XANES and EXAFS spectra after the desorption and after the re-absorption of hydrogen shows that Ti was not substituted into recombined $NaAlH_4$ again and that the formation of Al–Ti is irreversible. Regarding the cyclic properties of $NaAlH_4$ with Ti dopants, the hydrogen amount after the second time is approximately 20% less than that of the first, independent of the additive species. This indicates that parts of Al and Ti in $NaAlH_4$ are consumed to irreversibly form Al–Ti alloys. The hydrogenation–decomposition–desorption–recombination (HDDR) process is known as microstructure control technology [56]. This disproportionation reaction under a hydrogen atmosphere is sometimes utilized to obtain very fine crystalline grains or metastable phases. A similar reaction may occur in the first desorption/absorption reaction of the $NaAlH_4$–$TiCl_3$ system to prepare highly dispersed Al–Ti nanoparticles working as a catalyst for the subsequent hydrogen desorption/re-absorption.

DFT calculations revealed that the $NaAlH_4$ substituted by the amount of Ti in the AXS result is metastable. Because their configurational entropies are low (–0.019 to –0.33 kJ/mol atom for $x$ = 0.0078–0.0156), the stability of a single phase of the doped $NaAlH_4$ is not expected. However, the actual doping reaction includes the formation of other stable phases of NaCl and Al–Ti, as expressed in Equation (4). Gibbs energy calculations above the room temperature have shown the possibility that a high driving force with the formation of NaCl proceeds the addition reaction of $TiCl_3$ in $NaAlH_4$ [24]. On the other hand, Ti does not return to $NaAlH_4$ without the energy gain of the NaCl formation because the single phase of the doped $NaAlH_4$ is unstable. Therefore, it is considered that this doping reaction is irreversible with the consumption of the additives. Nevertheless, more precise calculations are expected to conclude the reaction path by determining the reaction products.

## 5. Conclusions

The irreversible disproportionation (decomposition/recombination) reaction hidden in the reversible hydrogen desorption/absorption reaction of $NaAlD(H)_4$–$TiCl_3$ was determined using pressure–composition isotherms, neutron/X-ray diffraction, anomalous X-ray scattering, X-ray absorption fine structure measurements, and DFT calculations. The reversible hydrogen desorption/re-absorption reactions of mechanically milled $NaAlD_4$ doped with 0.02 $TiCl_3$ were confirmed to be similar to those of the $NaAlH_4$–$TiCl_3$ system. It is suggested that Ti is substituted at the Al sites in $NaAlD(H)_4$ before the first hydrogen desorption and that the amount of substituted Ti decreases after hydrogen re-absorption, as confirmed by the refinement of the neutron/X-ray diffraction profiles. The anomalous X-ray scattering results revealed that $1.3 \pm 0.3\%$ of Ti was present at the Al site in $NaAlH_4$ before hydrogen desorption. Ti K-edge X-ray absorption fine structure curves indicated that mixed phases of Ti-substituted $NaAlH_4$ compound and Al–Ti alloys irreversibly changed to Al–Ti alloy, working as a catalyst for the reversible hydrogen desorption/absorption reaction. Highly dispersed Al–Ti nanoparticles prepared through the first decomposition of Ti-doped $NaAlH_4$ can possibly facilitate hydrogen storage. Therefore, the formation of Ti-substituted $NaAlH_4$ is an essential process for enhancing the catalytic effects of the Ti additives.

**Supplementary Materials:** The following are available online at https://www.mdpi.com/article/10.3390/app11188349/s1, Figure S1: X-ray powder diffraction profiles of $NaAlD_4$–0.02$TiCl_3$ before the desorption and after the re-absorption of deuterium. Figure S2: (a) Absorption edge energy and (b) relationship between the edge shift and the formal oxidation number of standard samples

(Ti: 0, TiO: 2, Ti$_2$O$_3$: 3, TiO$_2$: 4). Figure S3: Diffraction peaks of NaAlH$_4$–0.02TiCl$_3$ measured by using incident X-ray energies of (a) 4.6645 and (b) 4.9395 keV with background by linear function interpolation and integration range of $d$ = 4.55 ± 0.07 Å and 3.005 ± 0.035 Å for 101 and 112 of each Bragg peak, respectively. Table S1: Oxidation numbers of Ti in NaAlH$_4$–$x$TiCl$_3$ ($x$ = 0.005, 0.010, 0.015, 0.020, 0.040, and 0.060) before the desorption and after the re-absorption of hydrogen with standard samples of Ti, TiO, Ti$_2$O$_3$, TiO$_2$, TiH$_2$, TiCl$_3$, TiAl, Ti$_3$Al, and TiAl$_3$.

**Author Contributions:** Conceptualization, K.I. and T.O.; methodology, S.T.; investigation, F.F., H.O., T.H., T.K., H.A. (Hiroshi Arima), K.S. (Kazumasa Sugiyama), H.A. (Hitoshi Abe), H.K., K.S. (Kouji Sakaki), Y.N., A.M. and T.S.; writing—original draft preparation, K.I.; writing—review and editing, T.O. and S.-i.O.; supervision, T.O.; funding acquisition, K.I. and T.O. All authors have read and agreed to the published version of the manuscript.

**Funding:** This work was partially supported by JSPS KAKENHI (Grant Numbers 23686101, 24241034, 15K13810, 19K12650, JP18H05518 ("Hydrogenomics"), grants from the Inter-University Cooperative Research Program of the Institute for Materials Research, Tohoku University (Proposal Number 15K0110).

**Acknowledgments:** We acknowledge Hironori Nakao for his help in the anomalous X-ray scattering measurement. Synchrotron radiation X-ray diffraction experiments were performed using BL22XU at SPring-8 with the approval of the Japan Synchrotron Radiation Research Institute (JASRI) (Proposal No. 2014B3784) and under the Shared Use Program supported by JAEA Advanced Characterization Nanotechnology Platform under the remit of "Nanotechnology Platform" of MEXT Japan (Proposal No. JPMXP09A14AE0042). Neutron diffraction, anomalous X-ray scattering and X-ray absorption fine structure experiments were approved by Neutron Scattering Program Advisory Committee (Proposal Numbers 2014S06 and 2019S06), Photon Factory Program Advisory Committee (Proposal Numbers 2013PF-21, 2014T003), Multi-probe Program Advisory Committee (Proposal Number 2015MP003) of IMSS, KEK.

**Conflicts of Interest:** The authors declare no conflict of interest.

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
