# Peer review of "Generating Mechanism of Catalytic Effect for Hydrogen Absorption/Desorption Reactions in NaAlH4–TiCl3"

_applsci, doi:10.3390/app11188349_

Round 1
Reviewer 1 Report
There is a discrepancy between the nomination of the TiCl3-doped NaAlH4 in line 131, where it is written in the subindex (4-x). But in the following only the "4" is written in the subindex and the x is written in front of the Ti. Perhaps it would be better to explain this.
line 53: NaAlH4 is mentioned, but in the following you write about D Atoms and [AlD4] in line 56.
In equation (1)and (2) the arrow has to be replaced by two arrows pointing into both reaction directions.
line 82: .. alkoxide mechanical doped materials have low reversible capacities and release significant….
Equation (4): x < 0.1 is missing
line 90: ….Ti atoms or ions are generally favor substitution ...
line 129: Reference number [44] has to be correctly formatted.
line 158: ...with epoxy resin on the both sides.
line 163-164: He The (?) gas path was equipped around the sample and between the monitor and the sample holder filled with He.
line 180: ...a polyethylene bags in a globe glove box.
line 189: ...Debye-Waller factor
line 198-199: Meaning of the following sentence? "The k-mesh generated using the Monkhorst-Pack method with a spacing of less than 0.5 Å‒1".
line 216: "...the van’t Hoff plot shown in Fig. 1". In Fig.1 is no van-t Hoff plot, the van't Hoff plot is missingssing!
line 301: Simulation curves represent the Ti-substituted…….
line 335: Model B is slightly improved a little, but...
line 336-337: ... indicating lower Ti–Al coordination numbers or for Al–Ti phase than expected.
line 442: …. by determining of the reaction products.
line 518: The title is incomplete.
line 551: Title ?
line 568: Mistake in the title: NaAlH4 +xTiCl3
line 590: ...Bravais lattice determination....
line 605: ... Studies of Interatomice Distances in Halides and Chalecogenides.
line 607-608: ….hydrogenation-decomposition-desorption-recombination (HDDR)...
Reviewer 2 Report
Dear Authors!
I read your article with great interest.
The topic of the article is caused by the significant interest in the development of materials that can be used for solid-state hydrogen accumulation. The development of such materials requires a detailed understanding of the processes occurring both during their production and during operation. In the present work, a promising metal hydride NaAlH4 doped with TiCl3 which can significantly improve the performance of this compound has been investigated. The undoubted advantages of the work include a detailed literary review and the description of the utilized methods. An extensive set of diverse experimental techniques, together with quantum mechanical calculations and their analysis, allowed to draw a number of important conclusions about the atomic processes occurring during doping and re-absorption of hydrogen. The obtained results are of undoubted interest both from the point of view of understanding the physics of the interaction of hydrogen with solids and the for creating more advanced compounds for hydrogen storage.
The article is written clearly. I have no substantive remarks on the grammar and style of the English. I can definitely recommend the article for publication. I would make just several remarks to improve the understanding of the article:
- Lines 206-207: “residual minimization method direct inversion in the iterative subspace (RMM-DIIS)”.
Add “-”: “residual minimization method ‑ direct inversion in the iterative subspace (RMM-DIIS)”.
- Line 216: “was calculated by using the van’t Hoff plot shown in Fig. 1.”
Actually, Fig.1 represents P-C diagram. Van't Hoff diagram would be ln(p) vs 1/T.
- Liles 291, 454: “1.28±0.3%”
The last significant figure in any reported value should be in the same decimal place as the uncertainty.
Please change to:”1.3±0.3%”.
Also change “1.28” to “1.3” in lines 296, 331, 392, 406
- Lines 324-325: “radial distribution functions obtained from extended X-ray absorption fine structure (EXAFS) spectra using the three cluster models shown in Table 2.”
Insert the reference to Fig.5 b. For example:” radial distribution functions obtained from extended X-ray absorption fine structure (EXAFS) spectra (green line in Fig.5 (b)) using the three cluster models shown in Table 2.”
- Line 406: “So far, substitution of Ti with NaAlH4 has been suggested,”
The sentence "substitution of Ti with NaAlH4" implies that Ti is replaced by NaAlH4. While the authors obviously mean the opposite. Isn’t it?
